# On-Orbit Experimental Result of a Non-Deployable 430-MHz-Band Antenna Using a 1U CubeSat Structure

**Daisuke Nakayama** [1,*][iD]**, Takashi Yamauchi** [1][iD]**, Hirokazu Masui** [1]**, Sangkyun Kim** [1]**, Kazuhiro Toyoda** [1]**, Tharindu Lakmal Dayarathna Malmadayalage** [2]**, Mengu Cho** [1] **and the BIRDS-4 Project Team** [1,†]

1   Laboratory of Lean Satellite Enterprises and In-Orbit Experiments (LaSEINE), Kyushu Institute of Technology, Kitakyushu 8048550, Japan; yamauchi.takashi098@mail.kyutech.jp (T.Y.); masui@ele.kyutech.ac.jp (H.M.); kim.sangkyun571@mail.kyutech.jp (S.K.); toyoda@ele.kyutech.ac.jp (K.T.); cho.mengu801@mail.kyutech.jp (M.C.); birds4-project@kyutech-laseine.net

2   Arthur C. Clarke Institute for Modern Technologies, Moratuwa 10400, Sri Lanka; tharindu.dayarathna@accmt.ac.lk

*   Correspondence: nakayama.daisuke129@mail.kyutech.jp

†   Membership of the BIRDS-4 Project Team is provided in the Acknowledgments.

**Abstract:** 1U CubeSats often use the 430-MHz band for communication due to their size and power limitations, and half-wavelength dipole antennas are employed. A 430-MHz-band dipole antenna requires a deployable structure for a 1U CubeSat. However, a 1U CubeSat has a small volume margin for redundant systems, so the antenna deployment system can be a single point of failure. In this paper, the 1U CubeSat structure itself was used as an antenna. As a sub-mission of the BIRDS-4 project, three 1U CubeSats (GuaraniSat-1, Maya-2, and Tsuru) demonstrated this antenna structure. The results of the ground tests showed a maximum gain of −5.7 dBi with the flight model. These satellites were deployed from the International Space Station on 14 March 2021. Radio signals were alternately transmitted from the dipole antenna and the structure antenna onboard Tsuru for on-orbit demonstration on 15 December 2021, and the received signal strength on the ground was compared using RTL-SDR, SDR# and several codes. The ground station was able to receive both dipole and structure CW signals. The received power strength indicates that a gain of −8.1 dBi is being demonstrated with the structure antenna.

**Keywords:** 1U CubeSat; non-deployable antenna; on-orbit experiment

## 1. Introduction

The 1U CubeSat is one of the most popular lean satellite platforms. A lean satellite is a satellite that utilizes nontraditional, risk-taking development and management approaches with the aim to provide value of some kind to the customer at a low cost and without taking much time to realize the satellite mission [1]. A 1U CubeSat is about the size of a 10-cm cube. Many 1U CubeSats have carried 430-MHz amateur band transceivers and dipole or monopole antennas as their communication system due to limitations in volume and power generation capability. A 430-MHz-band antenna on a 1U CubeSat requires a deployable structure due to its wavelength. However, it is difficult to make such a deployable structure reliable. Particularly in the case of a 1U CubeSat, there is little volumetric margin for redundancy, so in many cases it is a single point of failure.

Jordi Puig-Suari et al. [2] prototyped a 1U CubeSat and developed the standard CubeSat deployer known as PolySat and P-POD (Poly-Picosatellite Orbital Deployer), respectively. It was one of the early CubeSat developments. It was equipped with 430-MHz band radio and four quarter-wavelength elements made by a spring ribbon. These elements were rolled under +Z panel before deployment from P-POD and held in place with monofilament. The monofilament was cut by nichrome wire after deployment and the antenna elements were deployed.

Yuichi Tsuda et al. [3] developed XI 1U CubeSat. The satellite was launched in 2002. The satellite was equipped with a dipole antenna of 430-MHz for downlink and a monopole antenna for 145-MHz for uplink. These antenna elements were also rolled on +X panel before deployment from P-POD and held with nylon wire that will be cut after deployed from P-POD by nichrome wire.

Marvin Martínez et al. [4] developed Quetzal-1 1U CubeSat. The satellite was launched in March 2020. It was deployed from J-SSOD (JEM Small Satellite Orbital Deployer) in April 2020. It was equipped a COTS (Commercial Off-The-Shelf) deployable antenna system manufactured by GOMSpace. The antenna was a turnstile antenna consisting of four monopole antennas. These four antenna elements were rigid rod type, and they had spring to deploy. Before being deployed from J-SSOD the elements were folded along the side of the CubeSat with a wire that was cut after being deployed from J-SSOD by heating resistors. The COTS antenna is now widely used by 1U CubeSats that employs a commercial CubeSat bus.

As shown in the three examples above, deployable antennas have been used for UHF in 1U CubeSat from the early days to the present. Suhala Abulgasem et al. [5] surveyed antennas used in the past CubeSats and compared 10 different antenna designs in the UHF band (300 MHz~1 GHz). Nine designs were deployable antennas and only one was a non-deployable antenna for 1.5U [6]. Non-deployable design for 1U was not addressed in Ref. [5].

Patch antennas have a low profile and the right weight characteristics and are candidates for non-deployable antennas; however, they can only operate efficiently at higher frequencies such as S-band within 10 cm × 10 cm. The power consumption of transceivers that handle such high frequencies tends to be high. There are few examples of such transceivers being installed in 1U CubeSats.

Meanwhile, there is some research about 430-MHz-band compact patch antennas that can fit on a 1U CubeSat. Md. Samsuzzaman et al. designed a 72-mm × 32-mm patch antenna [7] operating in the 430-MHz band and mounted it on a 1U CubeSat. A similar patch antenna was also designed independently for a 1U CubeSat at Tsukuba University and mounted on ITF-2 [8]. It supported not only the 430-MHz band but also the 145-MHz band. Both examples will occupy about half of a certain external panel of 1U CubeSat. This is a disadvantage in terms of power generation capacity.

In CubeSat designs, one has to make trade-off between the system reliability and the system performance. Non-deployable antenna avoids the satellite communication loss due to the deployment failure, making the satellite reliable. On the other hand, its performance is not as good as deployable antennas due to its small size. It barely satisfies the link budget. To have a larger antenna area without deploying, one generally needs to allocate a part of the satellite surface to the antenna, which consumes a precious surface for power generation. In the past, most 1U CubeSat developers chose the performance, i.e., the antenna gain, over the reliability. According to the Nanosatellite and CubeSat Database complied by Erik Kulu [9], there are 166 of 1U CubeSats that uses 430-MHz as the primary downlink frequency if we count the satellites launched up to November 2021. In the database, we can also see a photograph of each CubeSat. Judging from the photographs, we see only nine CubeSats and three projects use non-deployable antennas including BIRDS-1 [7], BIRDS-4 [10], and ITF-2 [8].

In this paper, a non-deployable antenna for the 430-MHz band using a different method from patch antennas is proposed. This uses the aluminum frame of a CubeSat as an antenna. This antenna does not require the deployable structure or occupy an external panel. This antenna was adopted as a sub-mission of the BIRDS-4 project [10] and was launched on three 1U CubeSats. This paper describes the results of the ground tests conducted before launching the satellites and the on-orbit demonstration conducted after launch.

The purpose of the present paper is to propose a novel non-deployable UHF antenna for 1U CubeSats and show the on-orbit results. The novelty is to use a part of the structure as an antenna that is specified by the CubeSat standard. We target 1U or 1.5U CubeSats where the resource such as volume or surface area available to the antenna is very scarce. The proposed antenna is reliable due to its nature of non-deployable antenna and consumes

the least resources in terms of volume and area. The paper is made of eight parts. The second part introduces a Loop Hentenna and the third part introduces the BIRDS-4 project. The fourth part introduces the HNT mission of BIRDS-4. The fifth part describes ground testing. The sixth part describes on-orbit demonstration. The seventh part presents analysis of the on-orbit results. The final part includes the conclusion and details of future plans.

## 2. Loop Hentenna

The "Hentenna" is an antenna developed by Japanese radio amateurs (such as JE1DEU [11]). The "Loop Hentenna" is a looped version of the "Hentenna". Figure 1 shows the structure of a Hentenna and a Loop Hentenna [12]. The black lines indicate conductors. Both antennas operate by feeding from the feed point indicated by the red circle in the figure.

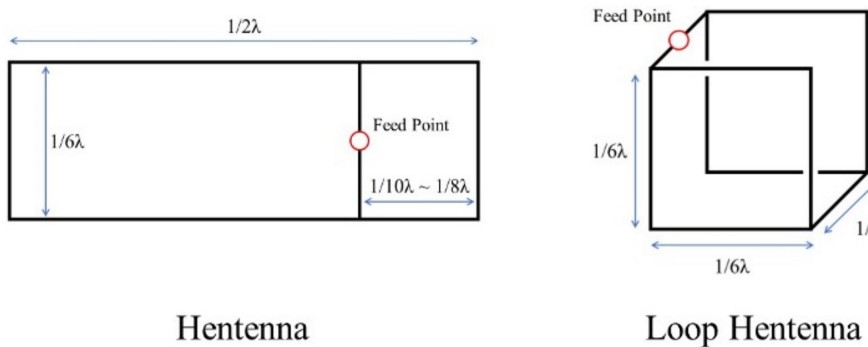

**Figure 1.** Structure of a Hentenna and Loop Hentenna.

A Hentenna has a planar structure. The impedance varies depending on the distance from the right end to the vertical bar where the feed point is located. A Loop Hentenna has a cubic shape with each side having 1/6 wavelength. The impedance is around 50 Ω. Both antennas have a figure of eight radiation pattern like a dipole antenna and the maximum gain is about 3 dBi. The polarization is linear polarization [11,12]. By setting one side of the Loop Hentenna to 10 cm, the frequency is 500 MHz. This shows that it is possible to build a 430-MHz antenna by applying the main structure of the 1U CubeSat.

*Comparison with Other Exsisting Designs*

Table 1 shows the comparison with existing non-deployable UHF antennas (300 MHz ~1 GHz) for 1U or 1.5U CubeSat. This table shows that the antenna proposed in this paper occupies smaller area than the other antennas because the rails and structural frame that are required for CubeSat anyway are converted to antennas. Our antenna requires the least amount of additional resource in terms of volume and surface area.

**Table 1.** Comparison of non-deployable UHF antennas (300 MHz~1 GHz) for 1U/1.5U CubeSat.

| Reference | Type (Target) | Frequency (MHz) | Gain (dBi) | Occupied Area |
|---|---|---|---|---|
| [7] | Slot (1.5U/3U) | 485/500 | 4 (simulated) | 2568 mm$^2$ (3 mm slot Total length: 856 mm) *1 |
| [8] | Patch (>1U) | 437.375 | 1.01 (measured) | 2720 mm$^2$ (80 × 34 × 2 mm) |
| [9] | Patch (>1U) | 437.525 MHz 145 | −23 (measured) *2 | ~2400 mm$^2$ (80 × 30 mm) |
| Proposed | Loop type (1U) | 437.375 | −5.7 (measured) | 1008 mm$^2$ (84 mm × 12 mm) *3 |

*1 Ignore the phasing line. *2 A 435 MHz band. *3 Ignore rail and structure surface area.

### 3. BIRDS-4 Project

The Joint Global Multi-National Birds or BIRDS project is a multinational CubeSat project led by the Kyushu Institute of Technology in Japan. The students must design, develop, and operate CubeSats within their master or doctor courses. The BIRDS-4 project is the 4th generation, and the mission statement of BIRDS-4 is building Paraguay's first satellite while improving the standardized bus system for future missions and give continuity to the satellite development of Japan and the Philippines, and previous missions from BIRDS-1, 2 and 3 [10]. The BIRDS-4 project began in November 2018 and the flight model was deployed in March 2021. Figure 2 shows the flight models of the BIRDS-4 satellites.

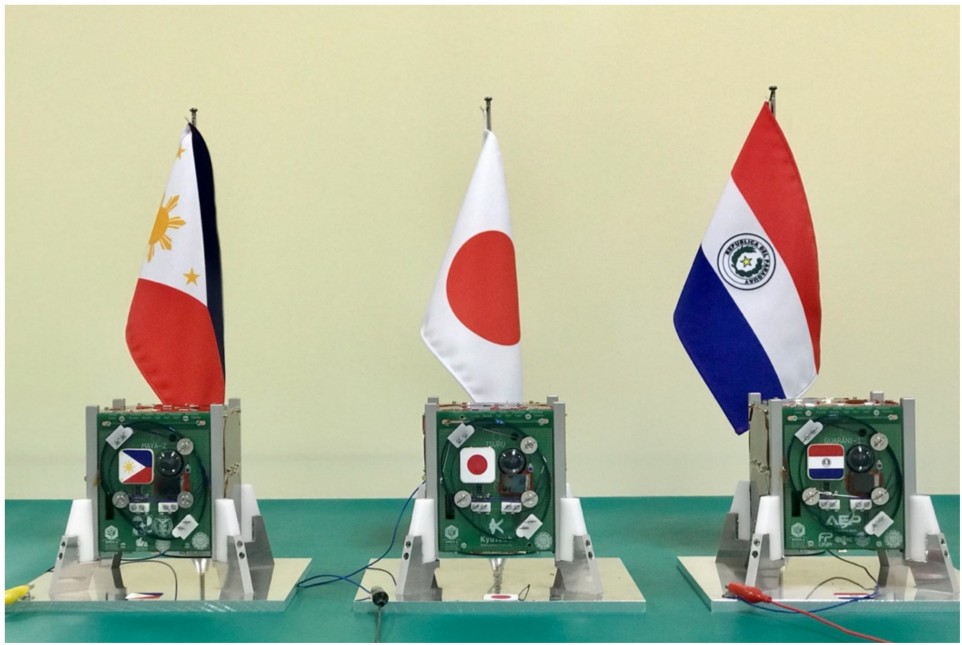

**Figure 2.** Flight models of BIRDS-4 satellites. (Left: Maya-2, Center: Tsuru, Right: GuaraniSat-1).

### 4. HNT Mission of the BIRDS-4 Project

The HNT mission is one of the BIRDS-4 missions. HNT stands for Hentenna. The mission objective is to demonstrate a Loop Hentenna. To achieve that, the CubeSat carried a Loop Hentenna and a dedicated CW (Continuous Wave) transmitter. Figure 3 shows the overall block diagram of BIRDS-4 satellite. The HNT mission works in coordination with the HNT mission block, MISSION BOSS, MAIN PIC, COM PIC, and UHF TRANCEIVER in the block diagram. BIRDS-4 has a complex internal connection due to its numerous missions. In this paper, only the operation of the parts related to the HNT mission will be briefly described.

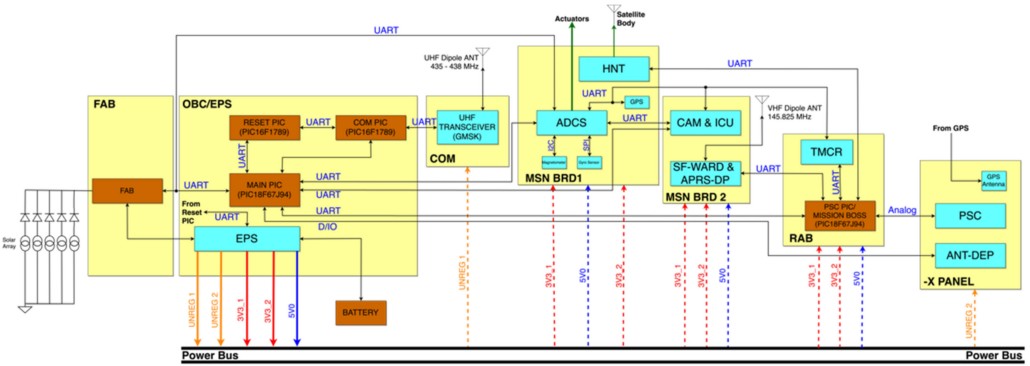

**Figure 3.** Overall block diagram of BIRDS-4.

The bus system always transmits a Morse beacon intermittently on CW using "UHF dipole ANT", as in Figure 3. During the HNT mission, the same kind of signal is transmitted from the Loop Hentenna, as shown as "Satellite Body" in Figure 3, during the off time of the beacon, as shown in Figure 4. By comparing the reception strength of these signals on the ground, we can estimate how much gain is achieved in space. The bus section uses a half-wavelength dipole antenna.

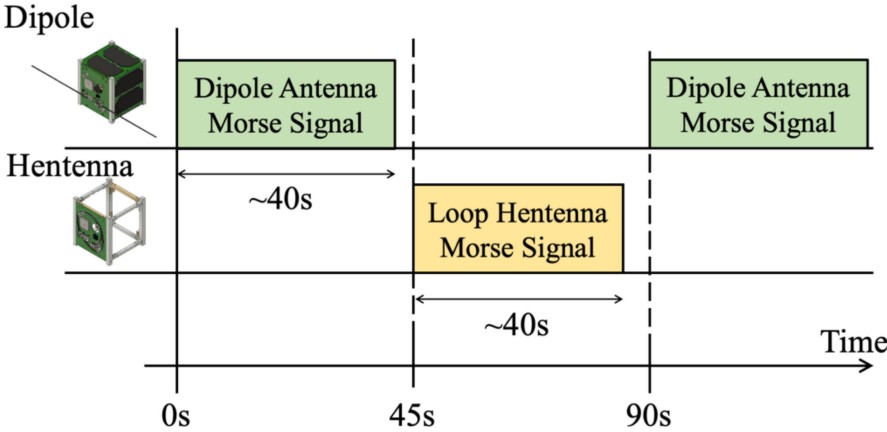

**Figure 4.** Order of signal transmission during the HNT mission.

Figure 5 shows the main internal structure. Part of the main structure has the aluminum used as a conductor replaced by PEEK (Poly Ether Ether Ketone), which is an insulator. This constitutes part of the Loop Hentenna structure, other than the horizontal element around the feed point. The feed point and the horizontal element are placed on the -X outer panel. Figure 6 shows a picture of the -X outer panel. It is a printed circuit board. The copper foil patterns connect the feed points to the rails of the CubeSat on either side. They are screwed to the ribs of the rails to connect electrically. In other words, the aluminum main frame and the -X panel act as a Loop Hentenna. In addition, an L-type matching circuit is built at the feed point to match the impedance of the antenna to 50 Ω.

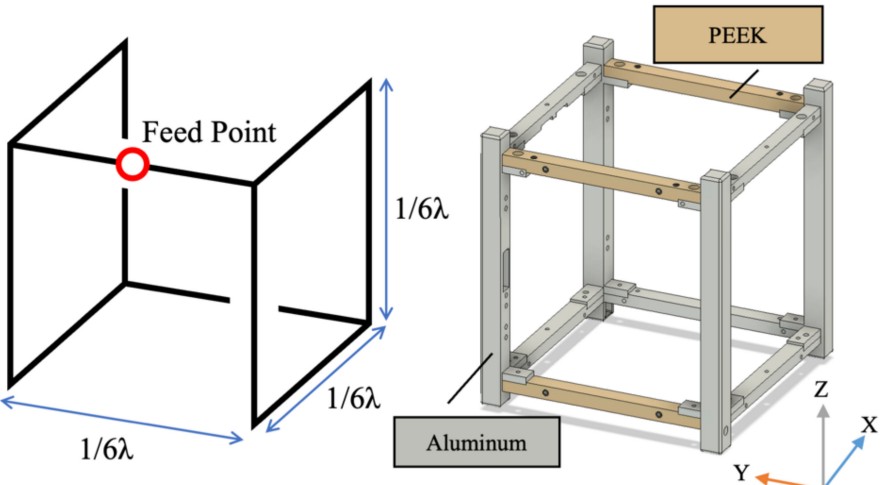

**Figure 5.** Loop Hentenna structure and BIRDS-4 main structure.

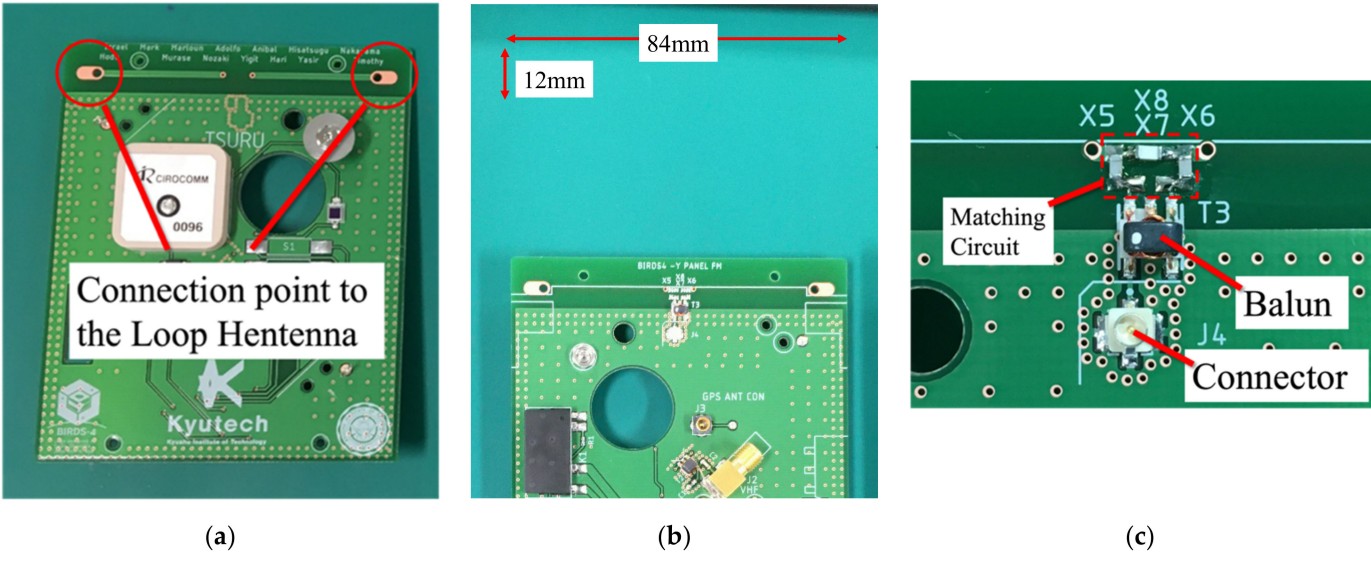

(**a**)            (**b**)            (**c**)

**Figure 6.** BIRDS-4 -X panel: (**a**) front; (**b**) back; (**c**) matching circuit and RF connector.

## 5. Testing on the Ground

The Loop Hentenna onboard the BIRDS-4 satellite was functionally tested before launch. This chapter describes the measurement of the reflection coefficient, the adjustment of the matching circuit and the measurement of the radiation pattern.

### 5.1. Reflection Coefficient

It is known that a Loop Hentenna with nothing inside has an input impedance of approximately 50 Ω, but when mounted on a CubeSat, it does not match 50 Ω because of the satellite components inside. Therefore, it is matched to 50 Ω using a matching circuit placed in the -X panel. This adjustment was carried out in the following way:

1. Make a through connection in the matching circuit; (L = 0 Ω, C = no implement in Figure 7)
2. Measure the input impedance by VNA (Rohde & Schwarz, ZNB 20);
3. Calculate the inductance L and capacitance C for the matching circuit;
4. Put L and C on the matching circuit;
5. Measure the input impedance by VNA;
6. Fine tune the values of L and C by looking at the Smith chart;
7. Repeat step 6 until the reflection coefficient is less than −10 dB.

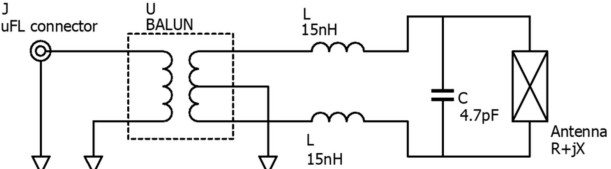

**Figure 7.** Schematic of the matching circuit. Balun: TC-1T+ made by Minicircuits. Inductor: L-15C15NJV4T made by Johanson Technology. Capacitor: 251R14S4R7BV4T made by Johanson Technology.

The results of the adjustments made to the FM of the Japanese satellite Tsuru are shown in Figures 7 and 8. The downlink frequency of BIRDS-4 is 437.375 MHz.

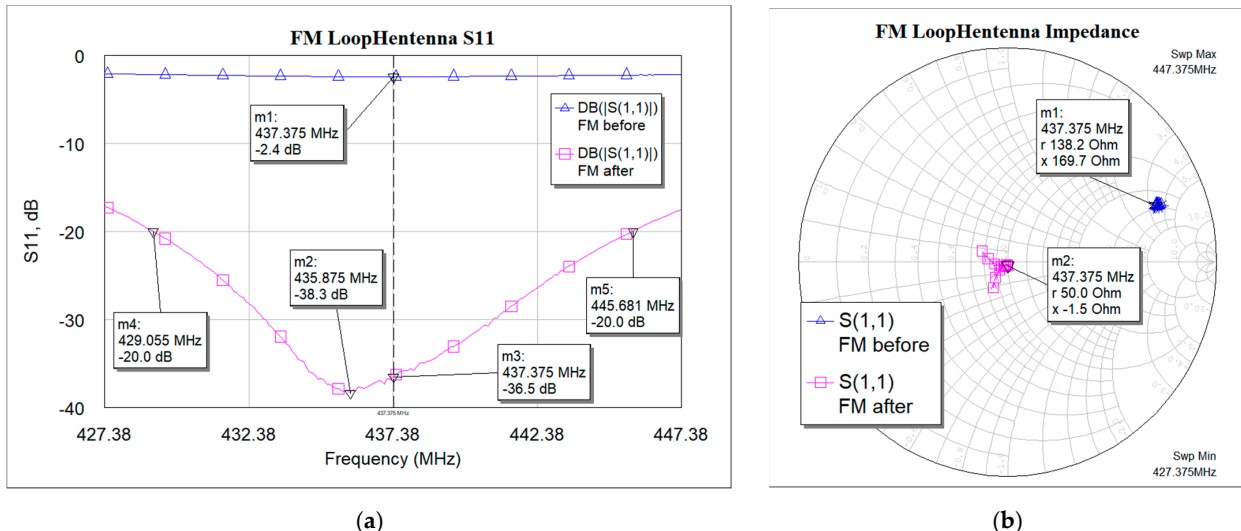

**Figure 8.** Result of return loss measuring: (**a**) return loss, (**b**) Smith chart.

The inductor and capacitor were L603 DS and S603DS, respectively, from Johanson Technology. The result of Z = 138.2 + j169.7 Ω at 437.375 MHz gives 44.31 nH and 3.85 pF as the inductor and capacitor values. We first chose 47 nH and 3.9 pF, but the reflection coefficient was not less than −10 dB at 437.375 MHz. Then we tuned the values checking the Smith Chart of VNA. Finally, we came to the values 15 nH and 4.7 pF. The difference from the initial estimate (44.31 nH and 3.85 pF) is probably due to parasitic inductance and capacitance of the wiring. The reflection coefficient was −2.4 dB before putting the matching circuit, but it was improved to −36.5 dB by adjusting the matching circuit.

### 5.2. Radiation Pattern

Radiation pattern tests were carried out using the anechoic chamber at the Kyushu Institute of Technology. Not only the ceiling and walls, but also the floor is covered with radio wave absorbers to simulate the space environment. Figure 9 shows the results of the radiation pattern measurement. It shows that the Loop Hentenna has a non-directional radiation pattern and a maximum gain of −5.7 dBi. The radiation pattern of the dipole antenna on BIRDS-4 was measured as well, with a maximum gain of 3.2 dBi.

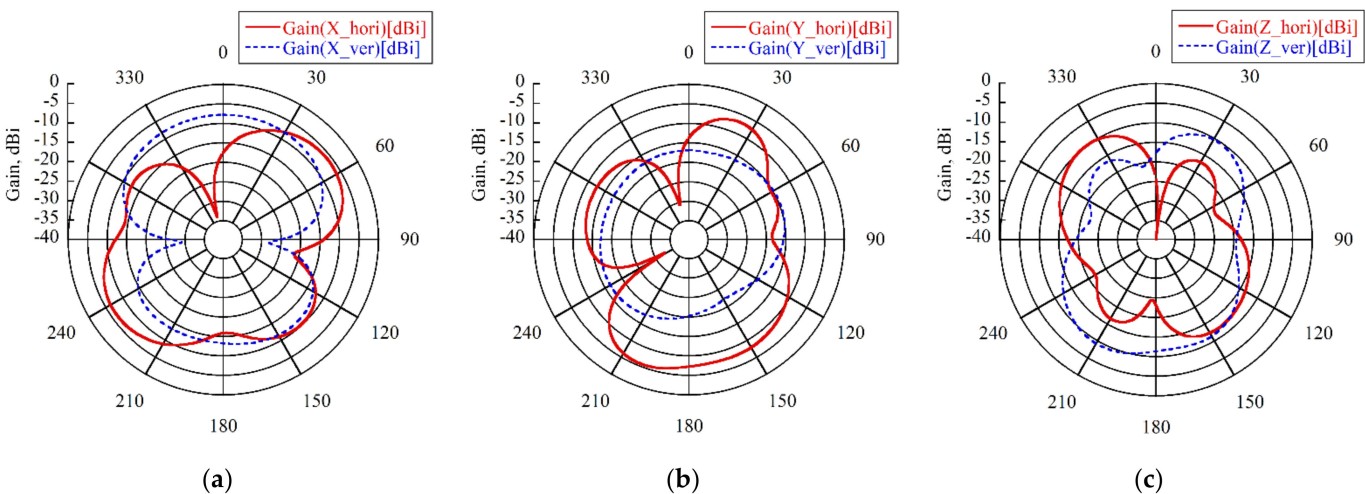

**Figure 9.** Radiation pattern measurement of the Loop Hentenna mounted on Tsuru (**a**) YZ plane, (**b**) ZX plane, (**c**) XY plane.

### 5.3. Transmitter Power

The transmitter for the Loop Hentenna is not shared with the bus system's transmitter to ensure that communication is not lost in the event of a mission failure. It has a dedicated CW transmitter. Figure 10 shows the transmitter for the HNT mission. The HNT mission transmitter shares the board with the ADCS (Attitude Determination and Control System) mission of BIRDS-4. The transmit power of the CW transmitters for the bus system and the Loop Hentenna were measured by spectrum analyzer with a 40-dB attenuator to compare them by radiating radio waves on orbit. Figure 11 shows the transmitting power of the Tsuru satellite. The CW transmitter for the bus system is part of the UHF communication board made by ADDNICS. Its transmitting power was 17.3 dBm. The CW transmitter for the HNT mission is based on an SX1278 chip from Semtech. Its transmitting power was 13.0 dBm. The difference was 4.3 dB.

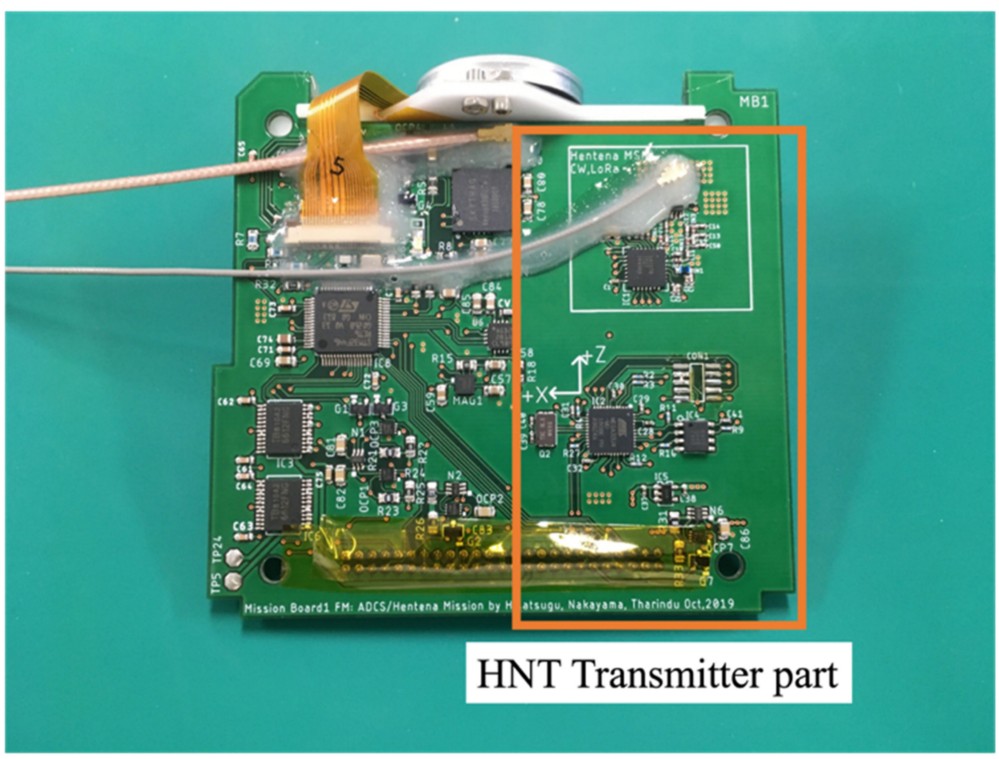

**Figure 10.** Transmitter for HNT mission.

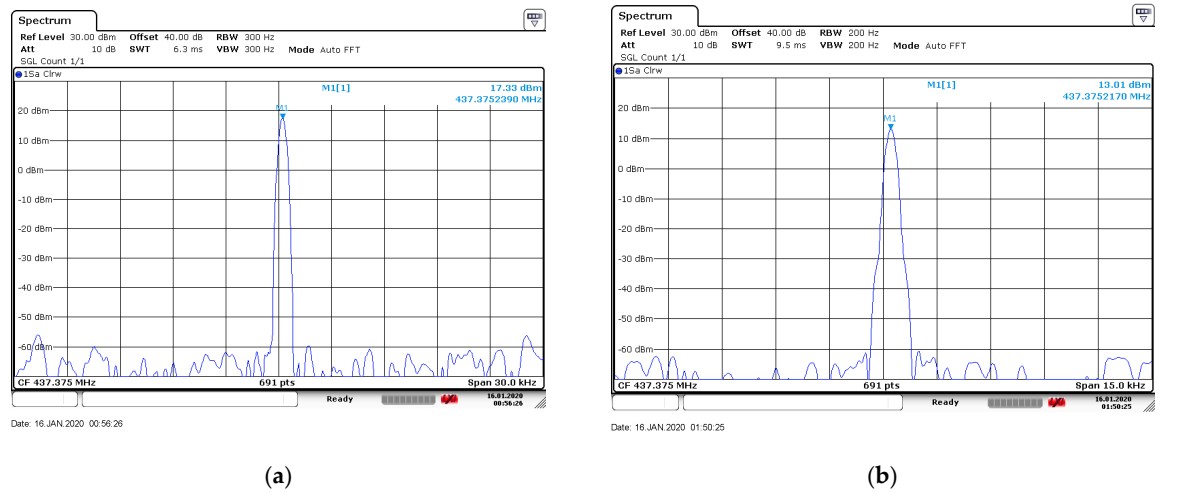

**Figure 11.** Transmitting power. (**a**) Bus system CW transmitter; (**b**) HNT mission transmitter (Tsuru).

### 5.4. Link Budget

Table 2 shows the link budget of HNT mission. The Loop Hentenna has a low gain, and the transmitter has a low output power. However, the ground station has a high-gain antenna with 22 dBi. Therefore, the link margin is positive when the satellite elevation is higher than 30 deg even if the link budget is calculated with conservative assumptions.

**Table 2.** The link budget calculation for HNT mission.

| Items | Value | Unit | Remark |
|---|---|---|---|
| Transmission Power | 13.0 | dBm | Measured Value |
| Line Loss (Satellite) | 1.0 | dB | |
| Antenna Gain (Loop Hentenna) | −5.7 | dBi | Measured Value |
| Pointing Loss | 10.0 | dB | |
| Polarization Loss | 3.0 | dB | Linar (Satellite) vs. Circular (Ground Station) |
| Ionization Loss | 1.0 | dB | |
| Atmosphic Loss | 1.0 | dB | |
| Rain Loss | 0.0 | dB | |
| Elevation | 30.0 | deg | |
| Range | 739.3 | km | 400 km altitude |
| Frequency | 437.375 | MHz | BIRDS-4 Hentenna Mission Beacon |
| Free Space Path Loss | 142.6 | dB | |
| Antenna Gain(Ground Station) | 22.0 | dBi | ref. M2 Antenna Systems, 436CP42UG 2 Stacks |
| Line Loss (Ground Station) | 3.0 | dB | |
| Receiving Signal | −132.3 | dBm | |
| Noise Temperature | 600 | K | |
| Bandwidth (BW) | 500 | Hz | |
| Noise Power within BW | −143.8 | dBm | |
| Signal Noise Ratio (SNR) | 11.5 | dB | |
| Required SNR | 10.0 | dB | |
| Link Margin | 1.5 | dB | |

## 6. On Orbit Demonstration

The satellites were launched on 22 February 2021 and deployed from the International Space Station on 14 March 2021. The HNT mission of the Tsuru satellite was turned on at 06:01 (the following are all in UTC), 15 December 2021, by command. The Tsuru satellite passed over the Kyushu Institute of Technology between 06:14 and 06:25. The maximum elevation was 68.5 deg at 06:20. During this pass, the ground station received signals from both the dipole antenna and the Loop Hentenna. Table 3 shows the specifications of the ground stations. Figure 12 shows an example of the receiving spectrum by RTL-SDR and SDR#. Please refer to "SDRSharp_20211215_061940Z_437375000Hz_IQ.wav" in Supplementary Materials for detailed data of this part. This shows the Loop Hentenna implemented on BIRDS-4 was working.

**Table 3.** Ground station specification for HNT mission at Kyushu Institute of Technology.

| Parameter | Value |
|---|---|
| Latitude (deg) | +33.8925 |
| Longitude (deg) | +130.8401 |
| Height | 50 m |
| Antenna | 22 element cross Yagi antenna × 2 |
| Antenna gain | 22.0 dBi |
| Receiver | RTL-SDR |
| Receiving software | AIRSPY SDR# v1.0.0.1831 |

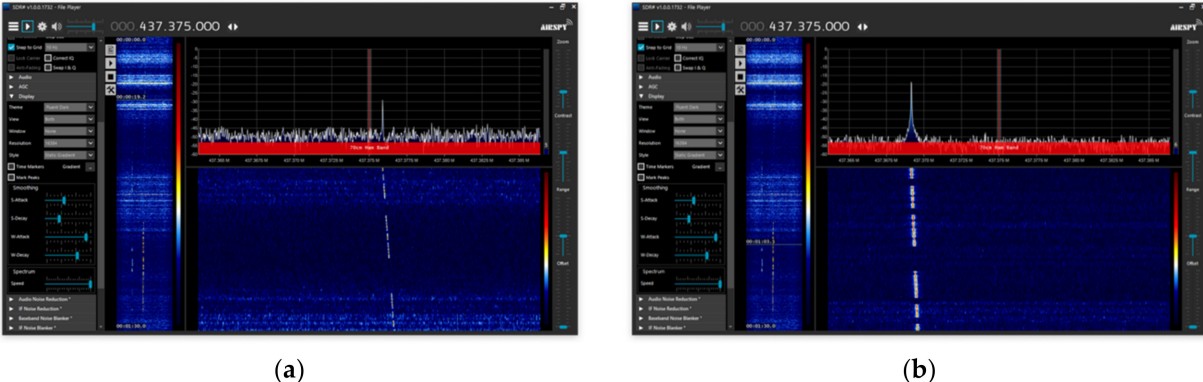

|        (**a**)        |        (**b**)        |

**Figure 12.** Receiving spectrum. (**a**) Loop Hentenna CW; (**b**) bus beacon CW.

## 7. Analysis from Recorded IQ Signal and Discussion

It is difficult to calculate the exact difference from the spectrum display of the SDR# because the CW beacon keeps keying and the signal changes its frequency every second with a Doppler shift. In addition, the distance from the ground station to the satellite changes every second, so the free space path loss also changes. Therefore, in this chapter the recorded IQ signal will be analyzed considering the change in distance.

### 7.1. FFT the IQ Signal

The IQ signal is a relatively large file and takes a long time to analyze, so the signal was cut out from the signal start from the Loop Hentenna (06:19:40, UTC) to the end of the CW beacon from the bus (06:21:10, UTC). The specifications of the IQ signal are shown in Table 4.

**Table 4.** IQ signal specifications.

| Parameter | Value |
|---|---|
| Sampling frequency | 250 kHz |
| Data start point | 15 December 2021<br>06:19:40 (UTC) |
| Data end point | 06:21:10 (UTC) |
| Samples | 22.5 M samples |
| (Duration) | (90 s) |
| ADC bits | 8 bits |

The FFT analysis uses 16,384 sampling data points obtained at a given time. Each FFT is done for 16,384 data points by shifting the starting data by 2048 data points, i.e., 8.192 msec. Figure 13 shows examples of the spectrum generated by FFT for given times. These figures show that the signal received at around 26.45 s is 14.6 dB lower than the signal at around 61.5 s. These peak frequencies become lower with time due to the Doppler shift. The peak frequency and intensity are the output at each FFT analysis. This analysis is performed for 22.5 million data points until 16,384 points data can no longer be extracted. After all, 10,979 FFT analyses were performed. Please refer to "fft_analysis.m" in Supplementary Materials for detailed code of this part. The specifications of the FFT analysis are listed in Table 5.

Figure 14 shows the strength history of the received signal from the Loop Hentenna and from the dipole antenna. Each block of CW messages is shown by a dotted-line square. CW messages from the Loop Hentenna are received before 50 s, CW messages from the bus system are received from 50 to 90 s. The parts of the signal that will be looked at in more detail later are indicated in the diagram. The Loop Hentenna sends a fixed message "BIRDS4 KYUTECH HNT MISSION 000". The dipole antenna sends housekeeping data that indicates the health of the bus system. It starts with "BIRDS4 JG6YMX", followed by 6 characters of message CW and 11 characters of housekeeping data [13].

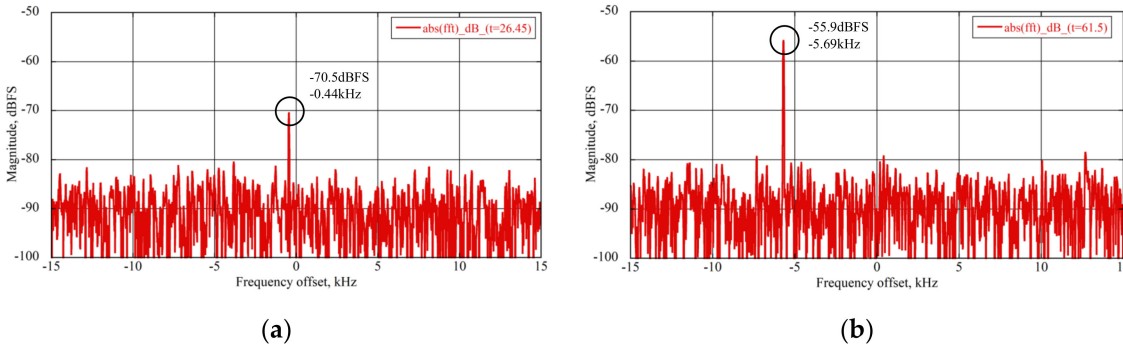

**Figure 13.** Examples of a spectrum generated by FFT. (**a**) Loop Hentenna CW (T = 26.45 s); (**b**) bus beacon CW (T = 61.5 s).

**Table 5.** FFT configuration.

| Parameter | Value |
|---|---|
| Shifting per FFT | 2048 samples |
| Samples used for FFT | 16,384 samples |
| (Duration) | (65.5 ms) |
| Window function | Flat-top |
| Total number of analyses | 10,979 |

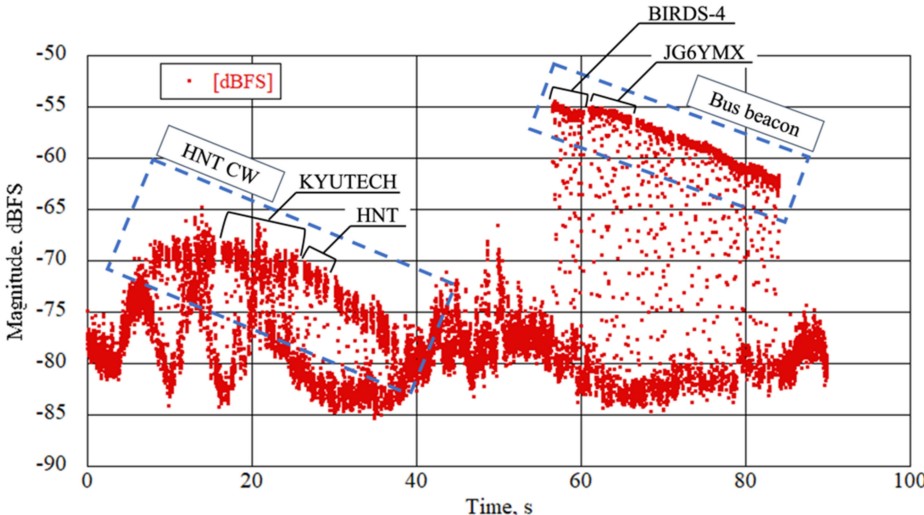

**Figure 14.** Elapsed seconds and peak intensity since 06:19:40 (UTC) on 15 December 2021.

As an example, an enlarged part of the graph from 26 to 28 s is shown in Figure 15. It is keyed "HNT" in Morse code, that is the pattern of " .... -. –". This string is not transmitted on a CW beacon signal from the bus system, so it can be confirmed that the signal is from Loop Hentenna.

An enlarged part of the graph from 60.5 to 66.5 s is shown in Figure 16. The signal is keyed to the Tsuru's callsign "JG6YMX". The HNT mission transmitter only transmits the specific message. It does not transmit the callsign. Only the transceiver of bus system transmits the callsign "JG6YMX" on a CW beacon signal. Therefore, it is concluded that the signal is from the bus system of the Tsuru satellite.

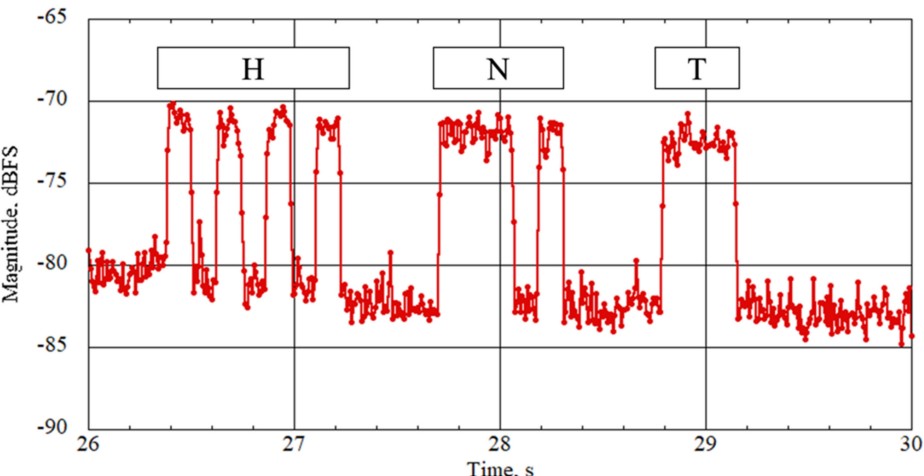

**Figure 15.** Received signal at elapsed time 26 to 30 s.

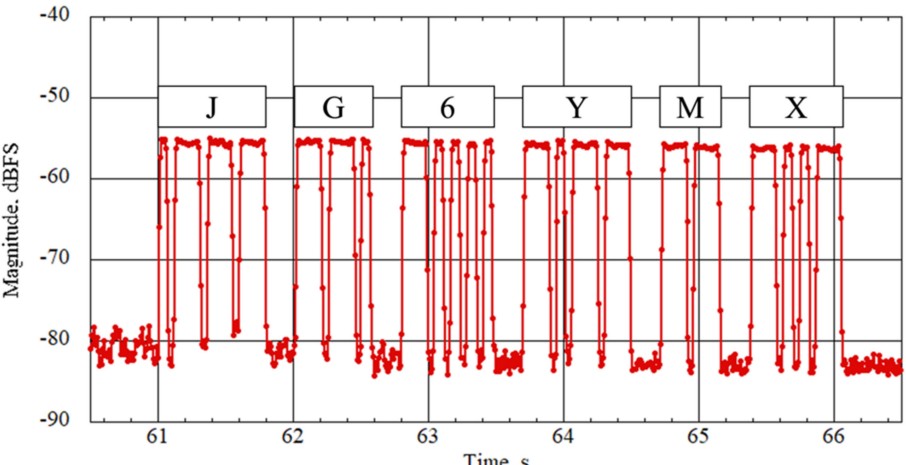

**Figure 16.** Received signal with elapsed time 60.5 to 66.5 s.

*7.2. Orbital Calculation from TLE and Correction of Signal Strength*

The latest TLEs of the time are available from space-track [14]. The contents of this are shown in Table 6. Python and the ephem library [15] were used to calculate the change in position and sight distance of the satellite at each signal arrival time. Please refer to "TLE2POS.py" in Supplementary Materials for detailed code of this part. The change in the sight distance over the period under analysis is shown in Figure 17. The distance varied from 425 to 658 km during reception. Since the free space loss is expressed by Equation (1), the respective Free Space Pass Loss (FSPL) varied from 137.9 to 141.6 dB, a change of 3.7 dB.

$$FSPL_{dB} = 20 \log\left(\frac{4\pi r}{\lambda}\right) \tag{1}$$

Figure 18 shows the normalization with respect to the most strongly received signal, considering the amount of change in free space loss. The highest signal of each beacon from Figure 18 is compared. A zoom view of the signal from Loop Hentenna "KYUTECH" is shown in Figure 19 and a zoomed-in view of the beacon "BIRDS4" from the bus system is shown in Figure 20. Focusing on the most strongly received letters, K and B, they have signal ratio strengths of −0.3 dB and −15.9 dB, respectively. The difference is 15.6 dB.

**Table 6.** Orbital element for calculation.

| Parameter | Value |
|---|---|
| Name | TSURU |
| NORAD ID | 47,927 |
| Epoch (UTC) | 14 December 2021 14:20:47 |
| Inclination | 51.639 deg |
| RA of A. node | 162.871 |
| Eccentricity | 0.0000359 |
| Argument of perigee | 276.418 |
| Revs per day | 15.56956963 |
| Mean anomaly | 83.677 |

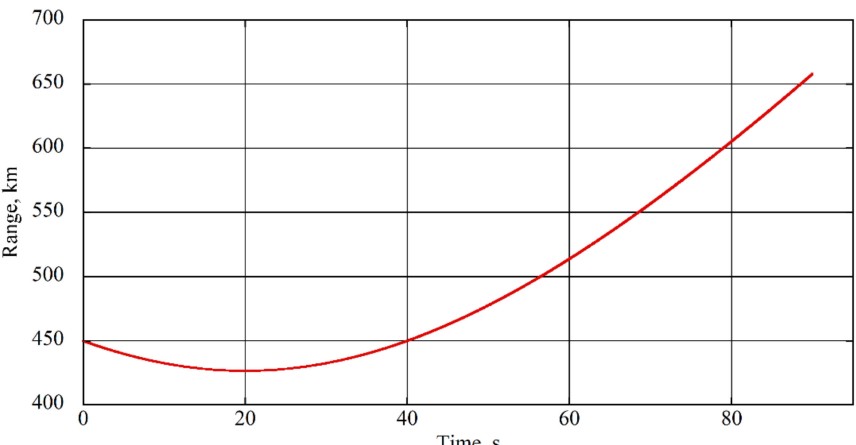

**Figure 17.** Elapsed seconds and the sight distance from the ground station to Tsuru.

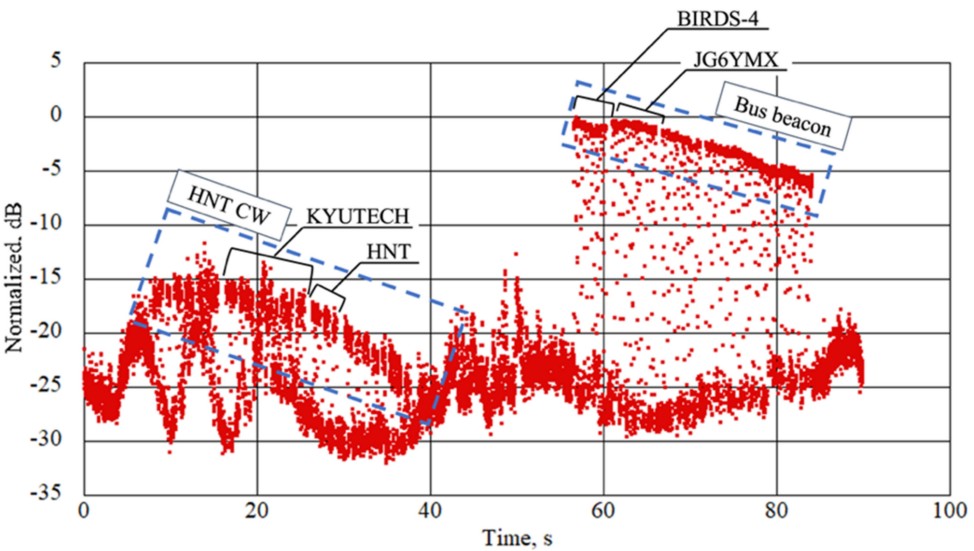

**Figure 18.** Elapsed seconds and normalized peak intensity since 06:19:40 (UTC) on 15 December 2021.

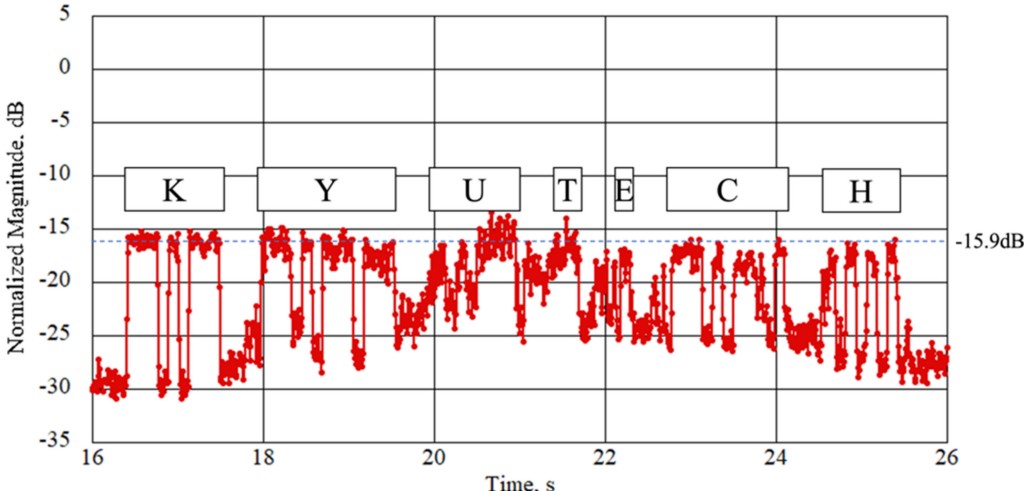

**Figure 19.** Received signal at elapsed time 16 to 26 s.

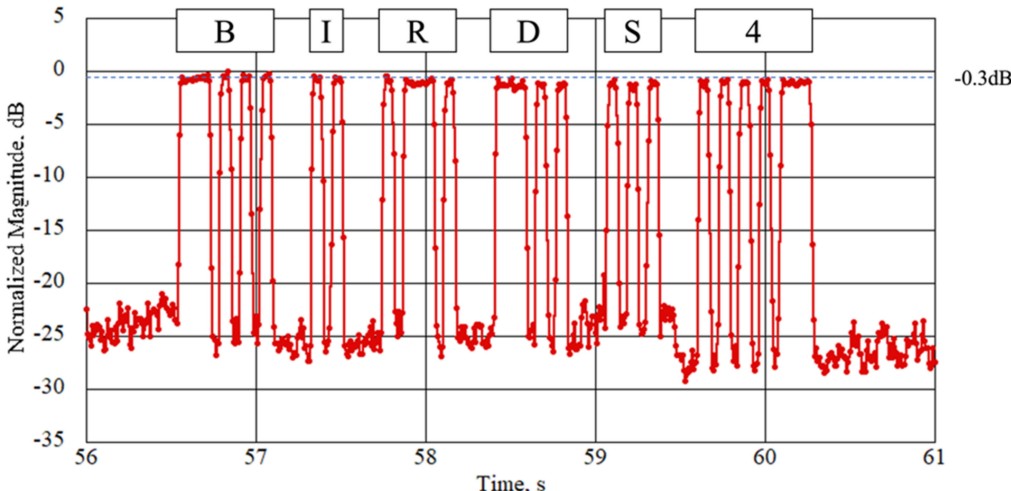

**Figure 20.** Received signal with elapsed time 56 to 61 s.

### 7.3. Estimates of the Gains Demonstrated in Space

Neglecting the change of attitude, the gain of the Loop Hentenna, $G_l$, can be obtained from the dipole antenna gain, $G_d$, and the respective transmitter outputs, $P_{txl}$, $P_{txd}$ and the difference of the receiving signal strength, $\Delta P_{rx}$, by the following equation:

$$G_l = G_d - P_{txl} + P_{txd} - \Delta P_{rx}. \tag{2}$$

As BIRDS-4 was not attitude controlled during the HNT mission, the attitude during the experiment is not known. Assuming that the gain of the dipole antenna is 3.2 dBi, as measured in the ground test, and all factors are the same except for the range distance and transmitter power, the Loop Hentenna gain $G_l$ is calculated as $-8.1$ dBi from $P_{txd} = 17.3$ dBm and $P_{txl} = 13.0$ dBm.

This value is 2.4 dB lower than the value obtained during the ground test, but this is due to many uncertainties, such as attitude. From the housekeeping data, the satellite rotational speed was typically 0 to 5 deg/s. It was confirmed that at least the Loop Hentenna is functioning as an antenna in space. The stable signal shows that the radiation pattern is non-directional.

The signal-to-noise ratio is at least good enough for listening and decoding the data. The number of missions that have been conducted is still low, so that needs to be increased to obtain more detailed data.

## 8. Conclusions

A non-deployable antenna called a Loop Hentenna, which uses the aluminum structure of a 1U CubeSat, was proposed. This antenna was mounted as a sub-mission on each of the three CubeSats of the BIRDS-4 project.

The Loop Hentenna measured maximum gain of $-5.7$ dBi before it was launched into orbit. The signals from the Loop Hentenna were successfully received after its launch. The signal was strong enough to decode its content. Analysis of the received signal showed that the antenna has a gain of $-8.1$ dBi.

As of January 2022, we expect that the satellite will stay in orbit for another year. We plan to do more on-orbit tests to collect data to characterize the antenna parameters.

## 9. Patents

Some of the results of this research have been filed as Japanese patents on 5 November 2019.

**Supplementary Materials:** The following supporting information can be downloaded at: https://www.mdpi.com/article/10.3390/electronics11071163/s1, File S1: fft_analysis.m: MATALB code for FFT Analysis; File S2: TLE2POS.py: Python code to calculate range distance to satellites and other information from TLEs; File S3: SDRSharp_20211215_061940Z_437375000Hz_IQ.wav: IQ signal recorded by SDR#.

**Author Contributions:** Conceptualization, D.N.; methodology, D.N.; software, D.N., T.L.D.M. and BIRDS-4 project; validation, D.N.; formal analysis, D.N.; investigation, D.N. and the BIRDS-4 Project Team; resources, T.Y., H.M., S.K. and M.C.; data curation, D.N.; writing—original draft preparation, D.N.; writing—review and editing, M.C.; visualization, D.N.; supervision, K.T. and M.C.; project administration, T.Y., H.M., S.K. and M.C.; funding acquisition, M.C. All authors have read and agreed to the published version of the manuscript.

**Funding:** This research was partially funded by JSPS Core-to-Core Program B: Asia-Africa Science Platforms (JPJSCCB20200005).

**Data Availability Statement:** Data are contained within the article or Supplementary Materials.

**Acknowledgments:** The development and operation of the BIRDS-4 project was based on the results of the three generations of projects up to and including BIRDS-3. The authors would like to express gratitude to the BIRDS-4 engineering team members in particular: Izrael Zenar Casople Bautista, Adolfo Javier Jara Cespedes, Anibal Antonio Mendoza Ruiz, Hari Ram Shrestha, Hiroki Hisatsugu, Hoda Awny Elmegharbel, Mark Angelo Cabrera Purio, Marloun Pelayo Sejera, Timothy Ivan Leong, Tomoaki Murase, Yasir Abbas, Yigit Cay, Yuma Nozaki, Mazaru Ariel Manabe Safi, Esteban Rafael Fretes Ruiz Díaz, and Akihiro Oboshi.

**Conflicts of Interest:** The authors declare no conflict of interest.

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
