# Peer review of "On-Orbit Experimental Result of a Non-Deployable 430-MHz-Band Antenna Using a 1U CubeSat Structure"

_electronics, doi:10.3390/electronics11071163_

Round 1

Reviewer 1 Report

This paper presents  non-deployable 430MHz band antenna  for 1U CubeSat communication system, which seems interesting in terms of technical point if view. However, the author should address the following comments to enhance the quality of the manuscript. 

  1. A comprehensive review should be done in introduction section.
  2. In line 257-//The Loop Hentenna measured maximum gain of -5.7 dBi before it was launched into the orbit//How the antenna will communicate with ground station with -5.7dB gain? What types of challenges faced to get data with this low gain? A Brief note  should be included in terms of project view.. 
  3. A comparison with existing techniques and antennas should be included.
  4. There should be a discussion on impedance matching of the antenna.
  5. If possible, the current distribution should be explained. 

Author Response

Thank you very much for your comments. We would like to answer each comment. The changes made as per the comment is marked in red in the revised manuscript. Please note that the numbers regarding page, figure, equation, table and so on are the ones used in the revised manuscript unless noted otherwise.

Comment 1: A comprehensive review should be done in introduction section.

 We added new references [2] and [3]. Ref [2] is a review for CubeSat antenna and introduces a non-deployable antenna for 1.5U CubeSat. We mentioned about the non-deployable antenna in Ref. [3].

Comment 2: In line 257-//The Loop Hentenna measured maximum gain of -5.7 dBi before it was launched into the orbit//How the antenna will communicate with ground station with -5.7dB gain? What types of challenges faced to get data with this low gain? A Brief note should be included in terms of project view.

The ground station has a high gain antenna of 22dBi, which is why we can communicate even with -5.7dBi in the satellite antenna. We made a new section 5.4. Link Budget was added as Table 2.

Comment 3: A comparison with existing techniques and antennas should be included.

 We added Table 1 to compare with other non-deployable antennas for 1U / 1.5U CubeSats.

Comment 4: There should be a discussion on impedance matching of the antenna.

 We added more description and discussion in Section 5.1.

Comment 5: If possible, the current distribution should be explained.

 Thank you for your good comment. We don’t have a good simulation result with internal components that explain antenna operation and current distribution so far.

Reviewer 2 Report

  • The novelty of this work is questionable. What is the contribution of this proposed design as compared to existing antenna designs? There are many non-deployable antennas that operates at different frequencies including 430 MHz and provide better performance.
  • The review of the literature is not thorough. Not all related work is well supported by relevant references, so the reader is not given an adequate background about the topic. This is important as it increases the accessibility to readers without strong background in the relevant area.
  • The paper is well structured and presented.
  • Most figures are not well presented.
  • The paper requires a proof reading. For example, see line 14.
  • Not all obtained results are well discussed and analyzed.
  • What is the polarization of the proposed antenna?
  • Figure 6, Line 130, replace “Return loss” by “reflection coefficient”. The correct definition of the Return Loss (RL) is: RL = 1/S11 (or RL_dB = -S11_dB); See the famous paper of Trevor Bird ( S. Bird, "Definition and Misuse of Return Loss [Report of the Transactions Editor-in-Chief]," in IEEE Antennas and Propagation Magazine, vol. 51, no. 2, pp. 166-167, April 2009, doi: 10.1109/MAP.2009.5162049.).

  • The design has not been compared with existing proposed designs for CubeSat that operates at the same frequency. Provide a table that compares the performance of your proposed antenna with other relevant existing antennas proposed for CubeSat. There are many other antenna designs operate at UHF and proposed for CubeSat. For example references [35], [70], [72] ..etc in the published review paper “ S. Abulgasem, F. Tubbal, R. Raad, P. I. Theoharis, S. Lu and S. Iranmanesh, "Antenna Designs for CubeSats: A Review," in IEEE Access, vol. 9, pp. 45289-45324, 2021
  • There are many other comments about the other sections of the paper, but the main concern is the contribution and Scientific of the presented design as mentioned above.

Author Response

Thank you very much for your comments. We would like to answer each comment. The changes made as per the comment is marked in red in the revised manuscript. Please note that the numbers regarding page, figure, equation, table and so on are the ones used in the revised manuscript unless noted otherwise.

Comment 1: The novelty of this work is questionable. What is the contribution of this proposed design as compared to existing antenna designs? There are many non-deployable antennas that operates at different frequencies including 430 MHz and provide better performance.

 We added the following sentences at the last paragraph of Introduction.

The novelty is to use a part of the structure as an antenna that is specified by the CubeSat standard. We target 1U or 1.5U CubeSats where the resource such as volume or surface area available to the antenna is very scarce. The proposed antenna is reliable due to its nature of non-deployable antenna and consumes the least resource in terms of volume and area.

Comment 2: The review of the literature is not thorough. Not all related work is well supported by relevant references, so the reader is not given an adequate background about the topic. This is important as it increases the accessibility to readers without strong background in the relevant area.

 Previously, we limited the scope of the literature review to 1U non-deployable antennas. We expanded to include all CubeSat non-deployable UHF antennas in the introduction citing the review article, Ref.[2], which was added in the revision.

Comment 3: The paper is well structured and presented. Most figures are not well presented.

We improved Figs.5 and 12 by adding some notes in the figures. We added some texts to explain Figs.12, 14, 15 more.

Comment 4: The paper requires a proof reading. For example, see line 14.

English of line 14 was corrected. Before submitting the original manuscript, we sent papers for a professional proof-reading service. A certificate of proof reading has been attached.

Comment 5: Not all obtained results are well discussed and analyzed.

We added some more texts in 7.1.

Comment 6: What is the polarization of the proposed antenna?

It is linear polarization.

Comment 7: Figure 6, Line 130, replace “Return loss” by “reflection coefficient”. The correct definition of the Return Loss (RL) is: RL = 1/S11 (or RL_dB = -S11_dB); See the famous paper of Trevor Bird ( S. Bird, "Definition and Misuse of Return Loss [Report of the Transactions Editor-in-Chief]," in IEEE Antennas and Propagation Magazine, vol. 51, no. 2, pp. 166-167, April 2009, doi: 10.1109/MAP.2009.5162049.).

 We replaced “Return loss” by “reflection coefficient”.

Comment 8: The design has not been compared with existing proposed designs for CubeSat that operates at the same frequency. Provide a table that compares the performance of your proposed antenna with other relevant existing antennas proposed for CubeSat. There are many other antenna designs operate at UHF and proposed for CubeSat. For example references [35], [70], [72] ..etc in the published review paper “ S. Abulgasem, F. Tubbal, R. Raad, P. I. Theoharis, S. Lu and S. Iranmanesh, "Antenna Designs for CubeSats: A Review," in IEEE Access, vol. 9, pp. 45289-45324, 2021 “

 Thank you for introducing the review paper. The paper mentions 10 UHF antenna designs. But only one is non-deployable and it is for 1.5U. We made a table for comparison (Table 1) and included the 1.5U design.

Comment 9: There are many other comments about the other sections of the paper, but the main concern is the contribution and Scientific of the presented design as mentioned above.

 Thank you very much for the good comments. Thanks to your comments, we believe the paper has improved.

Round 2

Reviewer 1 Report

Thanks for the reviewer response. The authors improved the manuscript. However, introduction section  should be improved by discussing more relevant lower UHF antennas of last five years. 

Reviewer 2 Report

Thanks for addressing most of the comments. However, the following comment has not been properly addressed. 

  • The review of the literature is not thorough. Not all related work is well supported by relevant references, so the reader is not given an adequate background about the topic. This is important as it increases the accessibility to readers without strong background in the relevant area.
